# Evaluation of the Impact of Activated Biochar-Manure Compost Pellet Fertilizer on Volatile Organic Compound Emissions and Heavy Metal Saturation

**DOI:** 10.3390/ijerph191912405

**Published:** 2022-09-29

**Authors:** Minsoo Kim, Changyoon Jeong, Minjeong Kim, Joohee Nam, Changki Shim, Joungdu Shin

**Affiliations:** 1Department of Environmental Engineering, University of Seoul, Seoul 02504, Korea; 2Red River Research Station, Agricultural Center, Louisiana State University, 262 Research Station Driver, Bossier City, LA 7112, USA; 3Organic Agriculture Division, National Institute of Agricultural Science, Wanjugun 55365, Korea; 4Eco-Friendly Environment & Microorganism Research Institute, Gyeonggi-Do Agricultural Research & Extension Services, Wanjugun 55365, Korea; 5Bio-Technology of Multidisciplinary Sciences, Co., 204 Ho JBTP R&D Center, 224 Wanjusan 6 Ro, Bondonggeup, Wanjugun 55315, Korea

**Keywords:** odor emissions, VOCs, heavy metals, activated biochar, pelletize, sustainability

## Abstract

For this experiment, pelletized activated biochar made of rice hullsor palm bark with swine manure compost was prepared to demonstrate the significant benefits of applying activated biochar-manure compost pellet fertilizer (ABMCP) inmitigating volatile organic compounds (VOCs), odor emission, and heavy metal saturation. Morphology and surface area analysis indicated that the activated rice hull biochar-manure compost pellet (ARP) had a significantly lower surface area, porous volume, and Fe content the activated palm biochar-manure compost pellet (APP). However, the ARP presented great potential to mitigate VOCs and odorant emissions. Our results indicated that the ARP reduced total reduced sulfur (TRS) and volatile fatty acids (VFAs) emissions by 69% and 93%, respectively. Heavy metals such as Pb, As, and Cd were not detected in the leachates fromthe ARP, APP, and swine manure compost. These results suggest that ABMCP can be a potential adsorbent to control VOCs and odorant emissions andpromote sustainable swine manure management and agricultural application.

## 1. Introduction

Swine manure compost contains a large amount of nitrogen (e.g., ammonia and amines), phosphorous (e.g., phosphoric acid), potassium (e.g., K_2_O), and various other trace substances, which are essential nutrients for crop growth. Therefore, applying manure compost to croplands increases soil fertility and crop productivity [1]. However, most of the swine manure in South Korea is not directly applied in farmland but converted into manure compost or liquid fertilizer [2]. In Japan, over 90% of swine manure is transformed into manure compost [3]. Composting is the most common stable and efficient treatment of animal waste including swine manure [4]. However, the composting of swine manure or its land application is generally not favored by the public due to the emission of volatile organic compounds (VOCs) associated with foul odors during the composting process and land application of composted products [4]. A variety of odor-causing substances such as ammonia, amines, organic acids such as propionic acid and butyric acid, and reduced sulfur compounds are the most common compounds released during the process of manure composting [5,6,7]. These odorous substances and VOCs affect the environment and create economic problems [8,9]. Especially severe incidences of VOCs or odor emission compounds have resulted in the closure of a composting facility and aban on manure composting [10]. More than 300 potential odorous VOCs are released from swine manure and facilities [11,12], and VOCs are a significant class of odorous compounds consisting of reduced sulfur compounds, volatile fatty acids (VFAs), phenols, indoles, amines, inorganic ammonia, and hydrogen sulfide. Although the detected VOC concentrations are under the odor detection threshold during the manure composting process, both organic and inorganic sulfides, indoles, and phenols are major compounds that contribute to odorous emissions [13].

Apart from VOCs and odor-causing substances, an additional limiting factor in using manure compost is the concentration of heavy metals [14]. Several studies presented that the long-term application of swine manure increased the concentrations of heavy metals such as cadmium, copper, and zinc in soil [15,16,17]. When fertilizers with swine manure compost were applied to soil, heavy metals accumulated in croplands, affecting soil health and food safety in cereal crop production [18]. Ultimately, the accumulation of heavy metals in agricultural soils not only pollutes the agroecosystem but may also have adverse effects on human health.

Biochar, a well-known highly porous organic material with a heterogenous surface, is produced via slow pyrolysis of biomass at relatively high temperatures without or with limited oxygen. Three main structural factors affect the adsorption characteristics of activated biochar, namely surface area, functional groups, and iron oxide content. After activation with KOH, the aromatic, hydrophobic, and non-polar properties of the biochar enhance to form an aromatized non-polar surface, and the aromatic properties are enhanced by increasing the pyrolysis temperature. The external surface of activated biochar is similar to the shape of a porous sponge with a mesoporous–microporous structure [19]. Several studies have used activated carbon to control the VOC emissions and odor removal from the air and improve water pollution [20]. Due to its large surface area and porosity driven by its physicochemical properties and surface chemistry, activated biochar has previously been adapted to mitigate VOCs and odorant emissions [21]. Biochar can decrease VOCs by more than 50% based on surface adsorption and reduce heavy metals such as Pb, Ni, Cu, Zn, As, Cr, and Cd [22]. Zhang et al. reported that biochar’s removal capacity of VOCs declined with increasing pyrolysis temperature, and both surface area and residual organic matter content play a key role in controlling the sorption capacities [23]. A few studies have documented that activated biochar can hold nutrients for extended periods and release nutrients slowly, controlling the transport of various pollutants in water and the atmosphere [24,25,26,27]. There is a limited number of studies on mitigating VOC emissions and heavy metal saturation (leaching) with the application of the pelletized activated biochar blended with manure in soils. Sessa et al. reported that pine-bark-based biochar significantly increased the adsorption of CO_2_ and toluene as model VOCs due to the derived microstructure, such as increased BET-specific surface [28]. Among the chemically activated biochar, the variety activated with K_2_CO_3_ exhibited the highest holding capacity. Activated biochar-manure compost pellet fertilizer (ABMCP) is defined as pelletized activated biochar based on rice hulls or palm bark combined with swine manure compost and supplemented with major nutrients.

In this study, we evaluated the impacts of two ABMCP on VOC emissions, various odorants, and heavy metals leaching soils. It is hypothesized that applying ABMCP mitigates the potential emissions of VOCs, TRS, and VFAs and reduces heavy metal contaminations in the soils.

## 2. Materials and Methods

### 2.1. Preparation of the ABMCP

In this study, we used different feedstocks to produce two varieties of biochar: activated rice hull biochar and palm biochar. Rice hull and swine manure compost was obtained from a cooperative agricultural union in Korea. Commercially available activated palm biochar was purchased from U-Gi Industry Co., GoChang, Jeon Buk, Republic of Korea. The activated rice hull biochar was prepared from rice hull biochar treated with 6 M KOH (1:2 ratios, 6 M KOH/biochar) through a pyrolysis process at 800 °C for 3 h in the absence of air. The materials were pelletized by mixing each activated biochar with swine manure compost in the 4:6 ratios reported earlier [29,30].

Fertilizers were mixed with the activated rice hull biochar-manure compost and the activated palm biochar-manure compost during the pelletizing process to prepare slow-release fertilizer. The characteristics of the materials are summarized in Table 1. The two types of ABMCP showed variation in their C, H, O, N, and S contents before and after mixing in swine manure compost. The N contents of the activated rice hull biochar-manure compost pellet (ARP) and the activated palm biochar-manure compost pellet (APP) exceeded 8–9% of total nitrogen content compared to rice hull biochar, swine manure compost, activated rice hull biochar, and activated palm biochar because of the supplemented nutrients.

### 2.2. Experimental Setup, Odor, and VOC Analysis

A diagram of the experimental setup to capture and concentrate the odorants from swine manure compost is provided in Figure 1. Swine manure compost and two types of ABMCP (20 g each) were placed individually in 0.2 L glass reactors. In each reactor, the moisture content was maintained at 30–35% (*w*/*w*). Air was flown at 16 mL min^−1^ through the headspace above samples in each glass reactor to release the odorants from the composting material. The air flushing was initiated one hour before taking each gas sample.

The VFAs emitted from each sample were extracted by exposing an 85 µm polyacrylate-coated solid-phase micro-extraction (SPME) fiber in the headspace of the reactor for 1 h. Then, the exposed SPME fibers were directly injected into the input port of a gas chromatography device equipped with a flame ionization detector (Shimadzu Scientific Instrument, GC-2010, Kyoto, Japan) for quantity analysis. The odorants extracted from the headspace using the SPME fiber and quantified cation were conducted via the gas chromatography device equipped with the flame ionized detector GC (FID). The dry air gas standards were produced by a permeation chamber in which base gas was flownover a tube with a known flow rate of 72 mL min^−1^ for the calibration. A detailed description of the SPME fiber calibration process was described by Kim et al. [31].

In addition, other odorous substances were analyzed by connecting the instrument to a flow meter. Sulfur-based odorous substances were monitored using the total reduced sulfur (TRS) analyzer (Teledyne Model 102E, Teledyne API, San Diego, CA, USA), and nitrogen oxide (N_2_O) was measured with the N_2_O analyzer (Teledyne Model T320U, Teledyne API, San Diego, CA, USA). VOC emissions were also detected with a portable VOC analyzer (ppbRAE 3000, RAE System, Inc., Sunnyvale, CA, USA). The laboratory temperature was maintained at 20 ± 2 °C during the experiment. The measured TRS, VOCs, and N_2_O values were evaluated by comparing the emission flux values (J) calculated from the following Equation (1).
Mass balance equation: Q_in_C_in_ + JA = Q_out_C_out_(1)
where Q_in_ and Q_out_ are the influent and effluent gas flow rates (mL min^−1^). C_in_ and C_out_ are the influent and effluent gas concentration (mL m^−3^). J is the emission flux (L min^−1^), and A is the surface area (cm^2^).

Since the influent gas (C_in_) used high-purity nitrogen, the measures of TRS, VOCs, and N_2_O were close to zero (C_in_ ≈ 0 mL m^−3^). Thus, the amount of influent and effluent gas was the same (Q_in_ = Q_out_). It was calculated by simplifying as shown in Equation (2).
J = Q_out_C_out_/A(2)

### 2.3. Biochar Characteristics and Elemental Composition

The surface morphology of the ABMCP was analyzed by the field emission scanning electron microscope (FE-SEM, Hitachi, SU8010, Tokyo, Japan). Biochar-manure compost pellets were held on an adhesive carbon tape on an aluminum stub followed by sputter coating with gold prior to viewing. Elemental composition analysis of the biochar powders and pellets was carried out via X-ray fluorescence spectroscopy (XRF, S2 Ranger, Bruker AXS GmbH, Karlsruhe, Germany). The N_2_adsorption–desorption isotherm measurements of the materials were analyzed with a Micromeritics sorption analyzer (Tristar II 3020, Micromeritics Instrument Corporation, Norcross, GA, USA) using N_2_ gas at −196.15 °C. Prior to the measurements, the samples were degassed at 299.85 °C under vacuum. The surface area and pore volume were calculated using the Brunauer–Emmett–Teller (BET) and the Barrett–Joyner–Halenda (BJH) equations, respectively.

### 2.4. Analysis of Heavy Metals in the Leachates

A cylindrical plastic reactor (1L) was used for the heavy metal elution test. The reactor has several 5 mm diameter holes to allow water to leach out from the bottom. Twenty grams of ABMCP and swine manure compost were weighed and put into the reactor, and 100 mL of deionized water was passed through the reactor once a week. The collected leachate from the reactor passed through a 0.2 µm syringe filter (Advantech Toyo Kaisha, Ltd., Cellulose Acetate Membrane Filter, Tokyo, Japan) to remove the suspended matter from leachate samples. The filtered water samples were treated with acid for further analysis. Concentrations of the heavy metals were measured using an inductively coupled plasma–atomic emission spectrometer (ICP-AES, ICPE-9800, Shimadzu, Kyoto, Japan). Ten different heavy metals (As, Ca, Cd, Cu, Fe, K, Ni, Pb, Si, and Zn) were measured.

### 2.5. Statistical Analysis of Data

The experiment had a completely randomized design with 3 replications. The data were subjected to analysis of variance (ANOVA) using SAS version 9.0 (SAS Institute Inc, Cary, NC, USA). The least significant difference (LSD, *p* < 0.05) test was applied to assess the differences among the means. All data were represented as standard deviation.

## 3. Results and Discussion

### 3.1. Characteristics of the ABMCP

The measured hydrogen/carbon (H/C) ratios were 0.105 in the ARP and 0.104 in the APP, respectively (Table 1). These biochar materials satisfied the criterion that H/C values must beless than 0.7 to be considered biochar according to standards provided by the International Biochar Initiative [32]. Aromatic ratios such as H/C and oxygen/carbon (O/C) ratios indicate the structural transformations [33] and surface hydrophilicity of biochar [34], such as the greater extent of functional groups containing O and H. H/C and O/C ratios are used in biochar classification to estimate the stability of biochar in soil, which is an international standard to calculate the carbon sequestration potential. The AMBCPs were suitable materials for soil carbon sequestration.

Surface area and porosity are essential parameters for the physical properties of biochar, and they play a crucial role in many biochar applications, especially for environmental remediation purposes. The surface morphology of the ABMCP made of rice hulls and palm bark are shown in Figure 2. There were apparent differences between the ARP and APP. The surface of the APP made of palm bark was more heterogeneous than the surface of the ARP from rice hulls. The ARP consisted of small particles and was smother, whereas the APP had an irregular shape. The surface structure of the ARP is more orderly, with smaller flakes than that of the APP. This difference is due to the different feeding stocks. Thus, the odorous substances and chemicals are exposed to relatively higher contact areas in the ARP compared to the APP.

The elemental XRF analysis confirmed that the carbon and potassium contents in the activated rice hull biochar increased up to 17.2% and 77.8%, respectively, compared with those of the rice hull biochar (Table 1 and Table 2). This may be attributed to the 6 M KOH treatment process for chemical activation. The elemental composition in ABMCP were Ca > K > Fe > Cl, which accounted for ~90% of the total elemental content (Table 2). The BET surface area of the APP was nearly 2.4 times higher compared with that of the ARP, indicating potentially less odor emission in the APP compared with the holding capacity of the ARP.

The N_2_ adsorption–desorption isotherms for the used materials are shown in Figure 3. The specific surface area, pore volume, and average pore size are summarized in Table 3. The surface area of the APP was seven times greater than that of the surface area of the ARP (Figure 3 and Table 3). In addition, the pore volume of activated palm biochar was nearly 3.5 times greater than that of the rice hull biochar observed in the FE-SEM image. In contrast to the above parameters, the APP’s average pore size was half the ARP’s. This indicates the larger surface area and implies that the APP shows higher potential to reduce odor emission compared to the ARP. However, the measured TRS and N_2_O emissions were not in agreement with the analyzed physical parameters (Figure 4). This may be due to the contribution of different physicochemical properties in the studied biochar.

The pores can be divided into nanopores (pore diameter < 0.9 nm), micropores (0.9 nm < pore diameter < 2 nm), and macropores (pore diameter > 50 nm). Generally, the surface area and total pore volume of biochar are commonly distributed within the ranges of 8 to 132 m^2^ g^−1^ and 0.016 to 0.083 cm^3^ g^−1^, respectively [35]. When a suitable precursor and preferable pyrolysis parameters are selected, the surface area and total pore volume of biochar can reach up to 490.8 m^2^ g^−1^ [36] and 0.25 cm^3^ g^−1^, respectively [37]. The measured surface areas from the activated rice hull biochar and palm biochar were 206.0 and 720.7 m^2^ g^−1^, and total pore volumes were 0.163 and 0.578 cm^3^ g^−1^, respectively (Table 3). This indicated that the ARP and APP were significantly improved in the distribution of pore sizes and morphological structure. The capability of the ARP and APP to adsorb VOCs and heavy metals was significantly enhanced via the activation process. The activated biochar enhanced cation exchange capacity, water holding capacity, and adsorption capacity, which increased due to the improved surface area and porosity [38].

### 3.2. Impacts of the ABMCP on Odor Emission of TRS and N_2_O

The average emission of TRS was 54.0 m^3^ m^−2^ h^−1^ in ARP treatment and 65.6 m^3^ m^−2^ h^−1^ in APP treatment compared with 159.0 m^3^ m^−2^ h^−1^ in the control (Figure 4). This suggests that the generation of sulfur-based odorous substances such as H_2_S and CS_2_ (carbon disulfide) from swine manure compost can be reduced by ARP and APP application. In addition, the reduction inN_2_O emission was more significant in the ARP treatment than in the APP treatment. The ARP also shows a high removal efficiency in N_2_O emission from swine manure compost. N_2_O emissions from the ARP and the APP treatments were 2.9 and 3.9 compared to 34.1 m^3^ m^−2^ h^−1^ from the control, with the average removal efficiency of 85% in ARP treatment and 79% in APP treatment (Figure 4).

Many studies on H_2_S adsorption using activated carbon (AC) have been performed because AC effectively decreases odor emissions under different environmental conditions [39,40]. Much research has been carried out on the use of biochar derived from agricultural byproducts in mitigating environmental pollutants [41,42]; however, limited information is available on biochar with H_2_S sorption and its binding mechanisms.

Adding 10% biochar reduced NH_3_, hydrogen sulfide (H_2_S), and total volatile organic compounds (TVOCs) contents by 20.04%, 16.18%, and 17.55%, respectively, and decreased the N loss rate by 8.27% compared to the control [43]. Previous studies have shown that biochar powder reduced the H_2_S and NH_3_ emissions compared to the pelletized form [44]. Plant-biomass-based biochar made from materials such as oak and coconut shells had considerably greater sorption capacity for dimethyl disulfide and dimethyl trisulfide than livestock-manure-based biochar [45]. Shang et al. also concluded that the removal capacity of H_2_S in rice-hull biochar was greater than in camphor biochar and bamboo biochar [46]. Similarly, our results showed that the removal efficiency of the ARP in total reduced sulfur (TRS) was 9% higher than the removal efficiency of the APP (60%) in this study.

### 3.3. Impacts of the ABMCP on VFAs Concentration and VOC Emission

Activated biochar with manure pellets is a potential cost-effective option to mitigate VOCs. The results of VOC emission showed that the average flux with ARP and APP treatments were 1233.7 and 1579.4 m^3^ m^−2^ h^−1^, respectively (Figure 5). This indicates that VOC emission with ARP treatment was about 21.9% less than that of the APP treatment. This indicates that the adsorption capacity of produced biochar entirely depends on the characteristics of feedstocks. Zhang et al. indicated that VOC sorption capacity from various types of biochar ranged from 5.58 to 91.2 mg g^−1^ [23]. Furthermore, it was reported that the effect of applied biochar in a vineyard was sustained for seven years according to measured soil functionality in a long-term field experiment [47].

The removal efficiency of volatile fatty acids (VFAs) concentration in ARP and APP treatments was93% and 88%, respectively (Figure 5). We note that the ARP seems to exhibit higher efficiency for the removal of VFAs compared to the removal efficiency of APP treatment. Duran et al. reported that the elevated biochar application rate reduced lower quantities of VFAs and other odors [48]. The 10% bamboo biochar (BB) amendment showed the stimulation of microbial activities that accelerated organic waste degradation and reduced VFAs and odor emission [23]. Awasthi et al. also presented that biochar application addition reduced the maximum VFA accumulations during wheat-straw biosolid composting [49].

### 3.4. Reduction in Heavy Metals from the Leachates through ABMCP Saturation

The application of ABMCP in cropland was used to evaluate the potential heavy metal contaminations in the agro-environmental system. As, Cd, and Pb were not detected in the leachates from swine manure compost, ARP, or APP treatment (Table 4). The high ratio of XRF analysis confirmed that Ca, K, Fe, and Si concentrations increased in the leachates from APP treatment compared to those from swine manure compost. On the other hand, micronutrients such as Ca and Fe concentrations in the leachates from ARP treatment were lower than those from APP treatment. However, these detected heavy metal contents were below the standard values in regulation provided in the Soil Environment Conservation Act [50] and Waste Control Act [51]. We conclude that ARP treatment demonstrated enhanced capabilities in controlling VOC emissions and heavy metal saturation.

## 4. Conclusions

In this study, odor and VOC emission were evaluated during the incubation of the ARP and APP. Although the ARP had lower surface area, pore volume, and sulfur content, it was highly effective in reducing odorants emission from swine manure compost and outperformed the APP. Overall, the average TRS and N_2_O flux from swine manure compost was 159.0 and 34.1 m^3^ m^−2^ h^−1^, respectively. The average removal efficiency of N_2_O was85% in the ARP and 79% in the APP treatments. The average VFA emission was 9 mg kg^−1^ from swine manure compost treatment. The TRS and VFA emissions were reduced by 69% and 93%, respectively, with ARP treatment, and by 60% and 88% with APP treatment. Heavy metals such as, Cd, and Pb were not detected in the leachates from swine manure compost and ABMCP treatments. These results confirm that ABMCP, especially that made of activated rice hull biochar, shows great potential for improving odor management in the animal manure compost industry and sustainable agricultural environment. The potential effects of ABMCP in the field, especially for the mitigation of VOCs, odor emission, and the retention of heavy metals in the soil, require further investigation.

## Figures and Tables

**Figure 1 ijerph-19-12405-f001:**
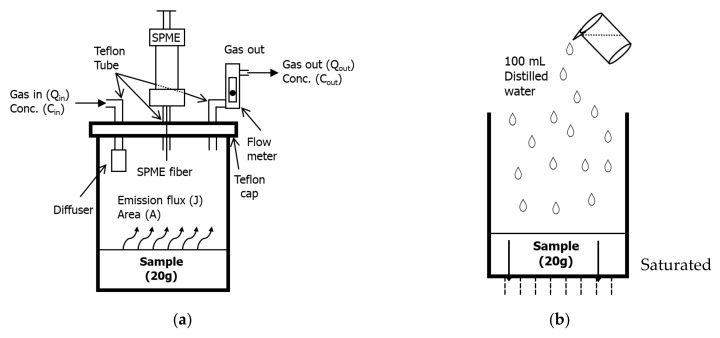
Schematic diagram of the set of the experiments: (**a**) reactor for VFAs, TRS, VOCs, and N_2_O analysis; (**b**) reactor for heavy metals in the leachates.

**Figure 2 ijerph-19-12405-f002:**
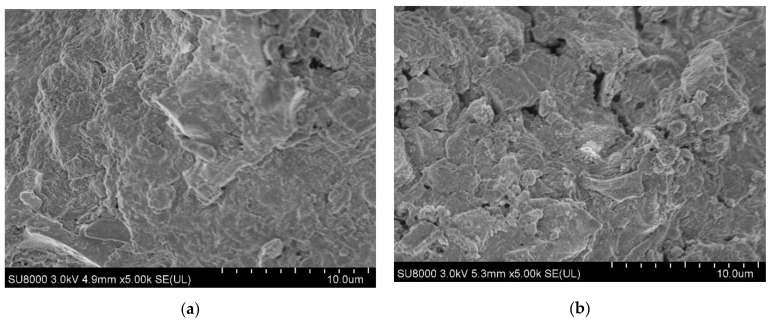
FE-SEM images of the activated biochar-manure compost pellets made of (**a**) rice hulls (ARP); (**b**) palm bark (APP).

**Figure 3 ijerph-19-12405-f003:**
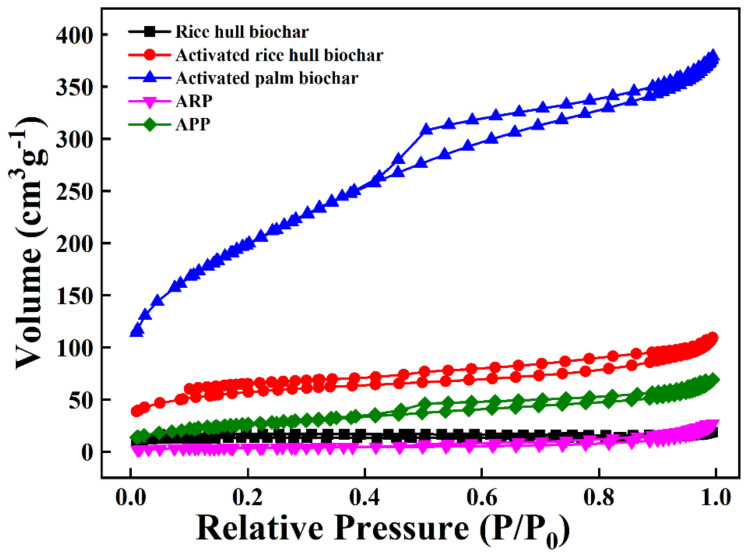
N_2_adsorption–desorption isotherm of the different types of used materials. ARP: activated rice hull biochar-manure compost pellet. APP: activated palm biochar-manure compost pellet.

**Figure 4 ijerph-19-12405-f004:**
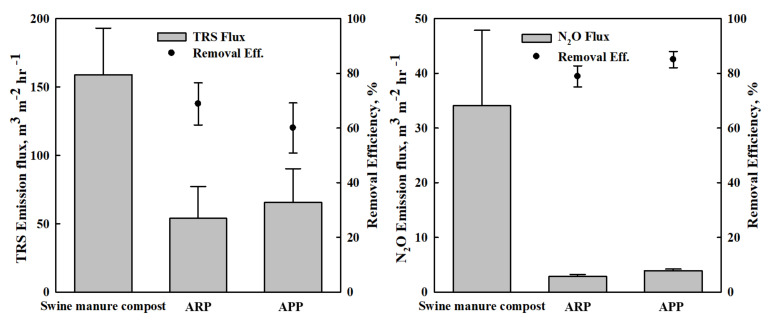
Total reduced sulfur (TRS) and N_2_O flux from swine manure compost and ABMCP. Error bars indicate standard error. ARP: activated rice hull biochar-manure compost pellet. APP: activated palm biochar-manure compost pellet.

**Figure 5 ijerph-19-12405-f005:**
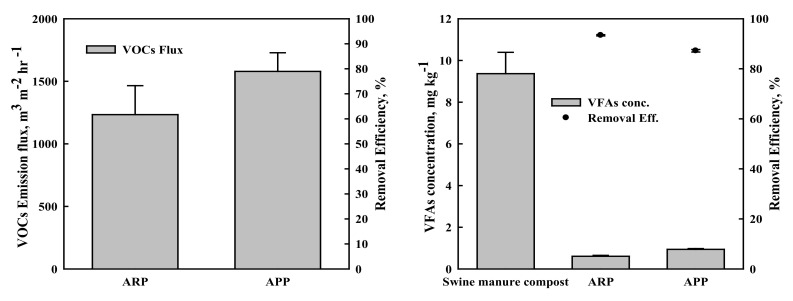
Volatile organic compound (VOC) flux and volatile fatty acids (VFAs) concentration from swine manure compost, ARP, and APP. Error bars indicate standard error. ARP: activated rice hull biochar-manure compost pellet. APP: activated palm biochar-manure compost pellet.

**Table 1 ijerph-19-12405-t001:** Characteristics of rice hull biochar, activated rice hull biochar, activated palm biochar, and their manure compost pellet materials.

Materials	Rice Hull Biochar (%)	Activated Rice Hull Biochar (%)	Activated Palm Biochar (%)	Swine Manure Compost (%)	ARP ^1^ (%)	APP ^2^ (%)
C	44.3	61.5	74.3	18.8	32.5	37.6
H	1.1	1.2	1.0	6.4	3.4	3.9
O	8.9	10.7	7.6	19.1	22.0	24.6
N	0.6	0.3	1.0	0.5	9.4	8.1
S	N.D. ^3^	N.D.	1.3	0.6	0.7	0.7
Others	45.1	26.3	14.8	54.6	32.0	25.1

^1^ ARP: activated rice hull biochar-manure compost pellet. ^2^ APP: activated palm biochar-manure compost pellet. ^3^ N.D.: not detected.

**Table 2 ijerph-19-12405-t002:** XRF result of rice hull biochar, activated rice hull biochar, activated palm biochar, and their manure compost pellet materials.

Materials	Rice Hull Biochar (%)	Activated Rice Hull Biochar (%)	Activated Palm Biochar (%)	ARP ^1^ (%)	APP ^2^ (%)
Calcium (Ca)	10.1	6.2	28.7	49.6	39.3
Potassium (K)	18.8	77.8	2.6	29.4	27.0
Iron (Fe)	1.5	1.9	35.9	7.0	16.9
Chlorine (Cl)	N.D.	N.D.	N.D.	8.2	9.9
Silicon (Si)	64.9	12.3	15.7	2.3	1.4
Sulfur (S)	0.7	N.D.	7.9	N.D.	1.2
Phosphorus (P)	0.6	N.D.	N.D.	1.1	1.2
Manganese (Mn)	3.0	1.2	1.7	N.D.	N.D.
Titanium (Ti)	N.D. ^3^	N.D.	4.6	N.D.	N.D.
Others	0.4	0.6	2.9	2.4	3.1

^1^ ARP: activated rice hull biochar-manure compost pellet. ^2^ APP: activated palm biochar-manure compost pellet. ^3^ N.D.: not detected.

**Table 3 ijerph-19-12405-t003:** BET result of rice hull biochar, activated rice hull biochar, activated palm biochar, and their manure-compost pellet materials.

Characteristics	Rice Hull Biochar	Activated Rice Hull Biochar	Activated Palm Biochar	ARP ^1^	APP ^2^
BET Surface Area (m^2^ g^−1^)	54.8 ± 1.07	206.0 ± 2.35	720.7 ± 2.32	12.8 ± 0.01	92.3 ± 0.36
Total pore volume (at BJH desorption) (cm^2^ g^−1^)	0.0307 ± 0.005	0.163 ± 0.011	0.578 ± 0.002	0.0454 ± 0.004	0.109 ± 0.02
Average pore size (at BJH desorption) (nm)	2.24 ± 0.03	3.17 ± 0.04	3.21 ± 0.01	10.6 ± 0.25	5.01 ± 0.15

^1^ ARP: activated rice hull biochar-manure compost pellet. ^2^ APP: activated palm biochar-manure compost pellet.

**Table 4 ijerph-19-12405-t004:** Heavy metals and light metals concentration (mean ± standard deviation, *n* = 3) in the leachates from swine manure compost and ABMCP.

Heavy Metals	Swine Manure Compost (mg L^−1^)	ARP ^1^ (mg L^−1^)	APP ^2^ (mg L^−1^)	Method Detection Limit (mg L^−1^)	Soil Environmental Conservation Act (mg kg^−1^)	Waste Management Act (mg L^−1^)
As	N.D. ^3^	N.D.	N.D.	0.08	25	1.5
Cd	N.D.	N.D.	N.D.	0.001	4	0.3
Cu	0.36 ± 0.15	0.30 ± 0.18	0.01 ± 0.01	0.002	150	3
Ni	0.009 ± 0.01	0.074 ± 0.03	0.004 ± 0.03	0.002	100	-
Pb	N.D.	N.D.	N.D.	0.01	200	3
Zn	1.37 ± 0.69	0.48 ± 0.29	0.59 ± 0.37	0.001	300	-
Fe	5.07 ± 2.60	3.17 ± 2.04	15.7 ± 10.6	0.001	-	-
Ca	17.9 ± 12.0	6.70 ± 3.42	31.2 ± 22.7	0.001	-	-
K	784 ± 53.6	1964 ± 102.4	2128 ± 161.9	0.002	-	-
Si	1.29 ± 0.48	2.04 ± 1.42	1.30 ± 0.73	0.002	-	-

^1^ ARP: activated rice hull biochar-manure compost pellet. ^2^ APP: activated palm biochar-manure compost pellet. ^3^ N.D.: not detected.

## Data Availability

Not applicable.

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
