# Peer review of "Evaluation of the Impact of Activated Biochar-Manure Compost Pellet Fertilizer on Volatile Organic Compound Emissions and Heavy Metal Saturation"

_ijerph, 2022, doi:10.3390/ijerph191912405_

Round 1

Reviewer 1 Report

Two little comments on pages 3 and 4.

Author Response

Thank you so much for your valuable comments.

Reviewer 2 Report

There are many researches about biochar as adsorbent for VOCs and heavy metal individually. The activated biochar manure pellet fertilizer synthesized presented  better absorption effect for the two kinds of pollution, which was the innovation of this paper. But the manuscript shows evidence of has or careless preparation, in particular important details have been omitted. I think the level of the article cannot meet the high requirements of the Int. J. Environ. Res. Public Health, and the review comments given are overhaul.

1. There is a lack of research on the relationship between the microstructure and adsorption characteristics of the material, and the analysis of materials is not convincing.

2. The author should give the structural test results, such as XRD, Raman, and so on.

3. The regeneration ability is an important performance for adsorbent, the relative research should be given.

Author Response

The author thanks to Reviewer 2 for valuable comments on this manuscript which has now been adjusted accordingly. 

Reviewer 3 Report

The manuscript presents a study of the short-term impact of pelletized activated biochars made of rice hull or palm barks with swine manure compost on mitigation of volatile organic compounds (VOCs), odor emission, and heavy metal saturation by demonstrating the significant benefits of applying activated biochar manure pellet fertilizer. I think that the manuscript presents interesting results and the research objectives have been achieved. The study could provide helpful information to enrich the knowledge on mitigating VOC emissions and heavy metal saturation under the application of the pelletized activated biochar and manure application in soils. Overall, the manuscript is worthy of publication. However, I have concerns that should be addressed before the paper could be published.

1. In the Introduction part, the authors introduced the necessity of mitigation of volatile organic compounds (VOCs), odor emission, and heavy metal saturation when composting the swine manure. It is better for the authors to supplement the latest related researches on the effects of application of the biochar to the manure, especially the similar other researches as the mitigation of volatile organic compounds (VOCs), odor emission, and heavy metal saturation when composting the swine manure.

2. In the Results and Discussions part, it is better for the authors to supplement the discussion on the duration of the mitigation of volatile organic compounds (VOCs), odor emission, and heavy metal saturation when composting the swine manure by applying the activated biochar manure pellet fertilizer. Moreover, it is better for the authors to extend and discuss the potential effects of the activated rice hull biochar manure pellets on the soil in the field, especially the mitigation of volatile organic compounds (VOCs), odor emission, and the retention of heavy metal in the soil compared with the control. In addition, the limitation of research and future research prospects may be mentioned in the part.

Author Response

(The authors gave the same response as above.)

Reviewer 4 Report

In the manuscript ID: ijerph-1901990 entitled “Evaluation of activated biochar manure pellet fertilizer on volatile organic compound emissions and heavy metal saturation”, the authors aim to demonstrate the significant benefits of applying activated biochar manure pellet fertilizer for mitigating VOCs, odour emissions and heavy metal saturation. The study methods are well described, although I strongly advise the authors to better specify N of data. Detailed comments are reported below.

- Line 72: Perhaps ABMP was defined in full only in the abstract: I recommend defining it for the first time in the text here, as well as TRS and VFAs.

- Table 1: I suggest the authors move Table 1 from section 2.1. to the results section (maybe at line 174).

- Figure 1: As done in Table 1, I suggest the authors to report the extended definitions of the acronyms used in the Figure.

- Line 168-169: I suggest citing here the reference to which the authors refer.

- Line 183: With respect to the morphological analysis, I ask myself how this was conducted: how many analyzes were performed? Has the procedure been standardized?

- Conclusions: I advise the authors to report here the advantages of the study, as well as the limitations of the work.

Author Response

(The authors gave the same response as above.)

Round 2

Reviewer 2 Report

I am satisfied with the author's response to my question